# *Salmonella* Brain Abscess in Sickle Cell Disease Patient: Case Report

**DOI:** 10.3390/reports7040107

**Published:** 2024-11-27

**Authors:** Felipe M. R. Monteiro, Ryan P. O’Boyle, Ruby R. Taylor, Danny L. John, Guilherme S. Piedade, Joacir G. Cordeiro

**Affiliations:** 1Department of Neurosurgery, Leonard M. Miller School of Medicine, University of Miami, Miami, FL 33136, USA; monteironeurosurgery@gmail.com (F.M.R.M.); gxs2230@miami.edu (G.S.P.); 2Medical Scientist Training Program, Leonard M. Miller School of Medicine, University of Miami, Miami, FL 33136, USA; 3Leonard M. Miller School of Medicine, University of Miami, Miami, FL 33136, USA

**Keywords:** *Salmonella*, sickle-cell disease, HbSS, brain abscess, brain infection, stereotactic drainage

## Abstract

**Background and Clinical Significance**: A brain abscess, defined as a localized intracranial infection that evolves into a purulent collection encased by a vascularized capsule, has higher prevalence among immunocompromised populations. Patients with sickle cell disease (SCD) are particularly vulnerable to bacterial infections due to their compromised immune systems, increasing their susceptibility to pathogens like *Salmonella*. While *Salmonella* is typically associated with gastroenteritis, osteomyelitis, and septicemia, its involvement in brain abscesses is exceedingly rare. There are few documented cases of *Salmonella* brain abscesses in the general population, and among patients with SCD, only one such case has been reported to date. In this report, we describe the second known case of a brain abscess caused by *Salmonella* infection in a patient with sickle cell disease, contributing to the limited literature on this rare and life-threatening condition. **Case Presentation**: A 32-year-old African American woman with sickle cell disease presented to the ER after a generalized seizure, reporting two weeks of worsening headaches, fevers, and left upper extremity weakness. Imaging revealed a right frontoparietal brain abscess, which was surgically drained, and cultures identified *Salmonella* enterica. After antibiotic treatment and a 23-day hospital stay, she was discharged. Four months later, she returned with another seizure during a sickle cell crisis, but follow-up MRI showed only minor scarring, and she was discharged on anticonvulsant therapy. **Conclusions**: This case emphasizes that *Salmonella* infections, though typically linked to osteomyelitis and sepsis, can also cause brain abscesses in immunocompromised patients like those with sickle cell disease. It highlights the need to consider infections alongside vascular causes in acute neurological cases and underscores the value of a multidisciplinary approach in managing such complex conditions.

## 1. Introduction and Clinical Significance

The *Salmonella* genus is divided into two species: *Salmonella enterica* and *Salmonella bongori*. *S. enterica* is further split into six subspecies, with *S. enterica* subspecies *enterica* being the most important, containing over 2500 serotypes. These serotypes are classified into two groups: typhoidal and non-typhoidal. Typhoidal serotypes, such as *Salmonella typhi* and *Salmonella paratyphi*, cause typhoid fever, while non-typhoidal serotypes are associated with foodborne illnesses, gastrointestinal infections, and conditions like osteomyelitis [1].

*Salmonella* is transmitted through the ingestion of contaminated food and water. After surviving the acidic environment of the stomach, the bacteria attach to the intestinal lining, where they invade and damage cells. The subsequent immune response produces symptoms of gastroenteritis [1]. In immunocompromised individuals, such as patients with sickle cell disease [2], non-typhoidal *Salmonella* can breach the intestinal barrier and enter the bloodstream, leading to bacteremia and potentially causing severe infections such as septicemia, endocarditis, osteomyelitis, or organ abscesses, including brain abscesses [3].

There have been only a few documented cases of *Salmonella* brain abscesses in the general population [4,5], and among patients with sickle cell disease, only one such case has been reported to date [5]. In this report, we describe the second known case of a brain abscess caused by *Salmonella* infection in a patient with sickle cell disease, contributing to the limited literature on this rare and life-threatening condition.

## 2. Case Presentation

A 32-year-old African American woman presented to the ER of an affiliated hospital in a postictal state following a witnessed generalized tonic–clonic seizure at home. She reported waking up with difficulty speaking and moving. Over the past two weeks, she had experienced progressively worsening right frontoparietal headaches, accompanied by fevers peaking at 106 °F, a mild cough, and dysuria. Despite these symptoms, she denied any significant focal infectious signs, intravenous drug use, recent travel, or gastrointestinal complaints. Additionally, for the past five days, she had been experiencing worsening left upper extremity (LUE) weakness, specifically with difficulty in distal hand grip. Her past medical history was notable for sickle cell disease (HbSS), multiple pain crises (the most recent occurring four weeks prior), splenectomy, and a left subclavian central venous port implanted 10 years ago.

On physical examination, the patient appeared drowsy, with noted LUE weakness (strength Grade IV in shoulder abduction, elbow flexion/extension, and Grade III in wrist flexion/extension and hand grip). Laboratory results showed severe anemia (Hb 5.2, Hct 16%), leukocytosis (WBC 15,800), thrombocytosis (platelets 852,000), and elevated inflammatory markers (ESR 126 mm/h, ferritin > 1000, CRP 3). CT imaging revealed a 3.2 cm × 3.2 cm hypodense mass with surrounding vasogenic edema in the right frontoparietal region, causing a 3.5 mm leftward midline shift (Figure 1a). MRI findings were consistent with a solid-cystic heterogeneous lesion with ring enhancement and hyperintensity on DWI, suggestive of a brain abscess (Figure 1b). She was transfused with one unit of packed red blood cells and transferred to our hospital the next day. She underwent frameless stereotactic evacuation of the brain abscess, with 20 cc of purulent drainage sent for cultures. Empiric antibiotic therapy with vancomycin, cefepime, metronidazole, and dexamethasone was initiated postoperatively.

Cultures from the brain abscess grew *Salmonella enterica* subspecies *enterica*, sensitive to Bactrim and resistant to levofloxacin. The urine culture showed *E. coli*, and blood cultures revealed *Bacillus cereus*, resistant to Bactrim but sensitive to levofloxacin, vancomycin, and penicillin. An echocardiogram and abdominopelvic CT scans were unremarkable, while a chest CT indicated thrombophlebitis in the subclavian venous catheter with septic emboli to the lungs. After the removal of the subclavian port, subsequent blood cultures were negative.

After a 23-day hospital stay, the patient was discharged on Keppra 500 mg every 12 h, Bactrim orally, and intravenous ceftriaxone for an additional four weeks. Four months later, the patient returned to the emergency department after experiencing another seizure, coinciding with a sickle cell crisis. Neurological examination was normal. An MRI showed near-normal cortical findings, with minor T2 hyperintensity (consistent with scarring) and no contrast enhancement. Following a ten-day hospital stay that ruled out infection—presumed to be reactive leukocytosis—the patient was discharged on 1500 mg of levetiracetam twice daily.

## 3. Discussion

Sickle cell disease (SCD) is a hereditary blood disorder caused by a mutation in the beta-globin gene, resulting in abnormal hemoglobin that polymerizes in capillary beds under low-oxygen conditions. This causes red blood cells to adopt a distorted sickle shape, disrupting normal blood flow and predisposing patients to ischemia and hemolysis [2]. The release of cell-free hemoglobin during hemolysis can trigger the activation of vascular endothelial cells, leading to increased expression of adhesion molecules and an excessive inflammatory response. The resulting endothelial damage, along with ischemia, contributes to multiple organ dysfunction [2,3].

Due to the ischemic-inflammatory nature of SCD pathogenesis, the most common neurological complications seen in hospitals are vascular-related, such as stroke, which is well-documented in the literature [2,6]. However, neurological complications related to infections are less often reported, underscoring the importance of cases like this one. In a case series of 70 pediatric patients with SCD admitted for neurological complications [7], 56 (78.6%) had strokes, while only 4 (5.7%) had brain abscesses.

A brain abscess is defined as a localized intracranial infection that begins as cerebritis and progresses into a purulent collection encased by a vascularized capsule [8]. The incidence of brain abscesses ranges from 0.3 to 1.3 per 100,000 individuals annually [8,9]. Brain abscesses may arise from contiguous spread from nearby infections (e.g., middle ear, mastoiditis), hematogenous dissemination, or trauma, such as open cranial fractures or complications following neurosurgical procedures. In the absence of nearby infections, trauma, or neurosurgical interventions [8,9,10], we hypothesize that hematogenous dissemination occurred due to disruption of the intestinal barrier and significant immune dysfunction caused by SCD.

Brain abscesses are primarily caused by bacterial infections, with the most common pathogens originating from oral cavity bacteria [8,9]. *Streptococcus anginosus* group, *Fusobacterium* spp., and *Aggregatibacter* spp. are frequently associated with dental or chronic ear infections. *Staphylococcus aureus* is commonly linked to brain abscesses after neurosurgical procedures or penetrating head trauma. Less frequently, enteric Gram-negative bacilli such as *Escherichia coli*, *Klebsiella* spp., and *Proteus* spp. can cause brain abscesses, particularly following bacteremia associated with gastrointestinal or respiratory infections. Protozoa like *Toxoplasma gondii* and *Mycobacterium tuberculosis* are also implicated in brain abscesses, particularly in immunocompromised individuals, highlighting the diversity of pathogens based on the patient’s underlying health and infection sources [8,9,11].

*Salmonella* is a far rarer cause of brain abscesses compared to the pathogens mentioned above. Mahapatra et al. [4] conducted a comprehensive review of 80 documented instances of focal intracranial *Salmonella* infections. Among these, 22 were classified as brain abscesses, 40 as subdural empyemas, and 18 as combinations of abscesses, empyemas, and epidural abscesses. Their analysis revealed that brain abscesses predominantly occurred in adults, while subdural empyemas were more common in pediatric populations [4].

Patients with SCD are highly susceptible to *Salmonella* infections [3]. The repeated blockage of blood vessels caused by sickled red blood cells leads to tissue necrosis, particularly in the intestines and bones, increasing permeability and allowing bacteria like *Salmonella* to enter the bloodstream more easily [2,3]. Additionally, SCD patients have compromised immune systems, primarily due to impaired spleen function or asplenia, which severely limits the body’s ability to filter out bacteria. Defects in the complement system, particularly reduced levels of C3 protein, further impair the ability to identify and clear *Salmonella*. Dysfunction in the reticuloendothelial system, including the liver and macrophages, reduces pathogen clearance from the bloodstream. These combined immune defects make it difficult for SCD patients to combat *Salmonella* infections, increasing their risk of severe complications [2,3,5].

The clinical presentation of brain abscesses is often nonspecific, making diagnosis reliant on imaging. Symptoms vary depending on the size and location of the abscess but may include headache, fever, focal neurological deficits, and altered mental status [8], as seen in our patient. Brain imaging is essential for accurate diagnosis, with MRI being the preferred method due to its superior sensitivity compared to CT [9]. MRI facilitates early detection of cerebritis, spread of inflammation, and identification of satellite lesions. On T1-weighted images, the abscess capsule typically appears as an isointense to mildly hyperintense rim, with gadolinium contrast enhancing the rim and surrounding edema. T2-weighted images show high signal intensity from surrounding edema, with an ill-defined hypointense rim [8,9]. Diffusion-weighted imaging (DWI) is particularly useful in differentiating abscesses from other ring-enhancing lesions, with a sensitivity of 95% and a specificity of 94% [12].

Treatment of brain abscesses involves a combination of antibiotic therapy and surgery. The choice of antibiotics, well-described in the literature, depends on the most likely causative organisms and is guided by culture results [9]. For community-acquired cases in non-HIV patients, an empirical regimen of a third-generation cephalosporin and metronidazole is a good option. The use of steroids remains controversial, though they may benefit patients with significant symptoms and perilesional edema, both of which were present in our case [11]. Surgery plays a crucial role, with stereotactic drainage generally preferred over craniotomy for resection because it is less invasive. However, in cases of superficial abscesses not involving eloquent brain regions, resection is preferable over drainage, especially when there is a concern for fungal or tuberculous infections or branching bacteria such as *Actinomyces* or *Nocardia* species [9]. Stereotactic drainage provides material for culture and helps relieve intracranial pressure and should be performed as soon as feasible. The typical treatment duration is 6 to 8 weeks, with progress monitored through clinical improvement, laboratory results, and imaging [8,11].

The prognosis for patients with brain abscesses has significantly improved over time. A systematic review of studies published between 1970 and 2013 found that the case-fatality rate decreased from 40% to 10%, while the rate of full recovery increased from 33% to 70% [10]. The most common sequelae are seizures, as observed in our patient. Seizures occur in up to 32% of cases and result from disruption of cortical integrity, leading to abnormal electrical activity in the brain. The healing process after a brain abscess often leads to scar tissue formation (gliosis), which can become a focus for abnormal electrical discharges, potentially resulting in chronic epilepsy [13].

## 4. Conclusions

This case highlights that *Salmonella* infections, though commonly associated with osteomyelitis, gastroenteritis, and sepsis, can also lead to brain abscesses, especially when immunity is compromised, as seen in patients with sickle cell disease. When evaluating acute neurological symptoms in these patients, it is essential to consider infectious causes alongside the more common ischemic-vascular etiologies. This case also underscores the importance of a comprehensive, multidisciplinary approach, drawing on knowledge from various subspecialties to understand how uncommon and complex conditions like this develop.

## Figures and Tables

**Figure 1 reports-07-00107-f001:**
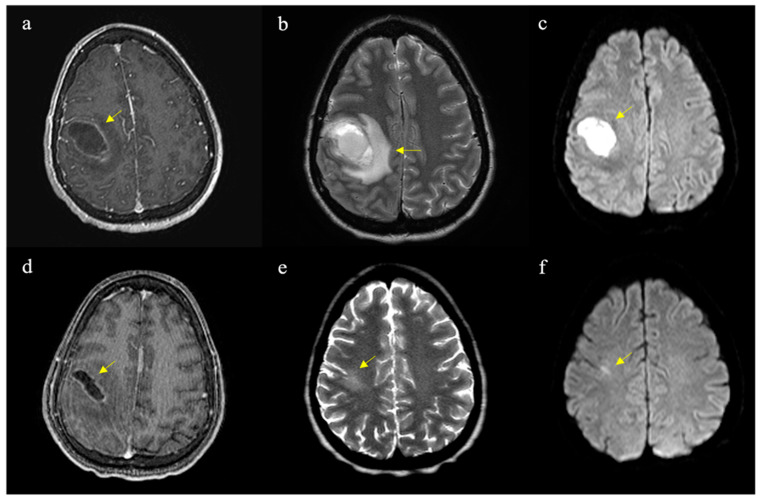
Axial MRI images pre- and post-operatively with different weightings. Typical abscess findings are indicated by arrows in each panel: (**a**) T1-weighted with contrast showing ring enhancement, (**b**) T2-weighted showing adjacent edema, (**c**) diffusion-weighted imaging (DWI) demonstrating marked hyperintensity, (**d**) 8-day post-operative T1-weighted image with contrast showing reduction in abscess volume compared to (**a**), and late (4-month) post-operative images showing slight residual hyperintensity in T2 (**e**) and no water restriction on the DWI (**f**) sequences.

## Data Availability

The data presented in this study are available on request from the corresponding author. The data are not publicly available due to privacy restrictions.

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
