# Peer review of "Salmonella Brain Abscess in Sickle Cell Disease Patient: Case Report"

_reports, 2024, doi:10.3390/reports7040107_

Round 1

Reviewer 1 Report

Comments and Suggestions for Authors

The authors described a case report of successful treatment of salmonella brain abscess in patient with sickle cell disease.

The introduction is short and concise. The case itself is well presented with a clear description of diagnostic procedures and treatment of the abscess. The figure 1 is informative.

In the discussion, the authors describe more broadly the sickle cell disease and its connection with infections. The describe the possible clinical course of brain abscesses.

Overall, the case is nicely written and describes a rare occurrence of salmonella brain abscess in sickle cell disease. The authors should describe more in detail (in the discussion) also the total surgical excision of an abscess (removal of pus and capsule of the abscess) and its advantages and disadvantages compared to simple abscess aspiration. After these minor corrections, the article is suitable for publishing in the journal.

Author Response

Comments 1: 

The authors described a case report of successful treatment of salmonella brain abscess in patient with sickle cell disease.

The introduction is short and concise. The case itself is well presented with a clear description of diagnostic procedures and treatment of the abscess. The figure 1 is informative.

In the discussion, the authors describe more broadly the sickle cell disease and its connection with infections. The describe the possible clinical course of brain abscesses.

Overall, the case is nicely written and describes a rare occurrence of salmonella brain abscess in sickle cell disease. The authors should describe more in detail (in the discussion) also the total surgical excision of an abscess (removal of pus and capsule of the abscess) and its advantages and disadvantages compared to simple abscess aspiration. After these minor corrections, the article is suitable for publishing in the journal.

Response 1:

Thank you for your comments! We have updated the manuscript with an explanation of the indications for abscess excision vs drainage. These changes are marked in red from line 175 to line 178 in the updated manuscript. We added the following (in red): "Surgery plays a crucial role, with stereotactic drainage generally preferred over craniotomy for resection because it is less invasive. However, in cases of superficial abscesses not involving eloquent brain regions, resection is preferable over drainage, especially when there is a concern for fungal or tuberculous infections or branching bacteria such as Actinomyces or Nocardia species [9]. "

Reviewer 2 Report

Comments and Suggestions for Authors

I compliment the authors on a well written case report dealing with Salmonella brain abscess in a patient with sickle cell disease

Author Response

Comments 1: I compliment the authors on a well written case report dealing with Salmonella brain abscess in a patient with sickle cell disease.

Response 1: Thank you for your kind comment!

Reviewer 3 Report

Comments and Suggestions for Authors

This is a well-written case report. They described a Salmonella brain abscess in Sickle Cell disease, and as stated in the study, this is the second description in the literature. They believe the mechanism is hematogenous spread. They provided their management strategies, as well as post-operative and pre-operative imaging for their case.
I believe this is a useful case report for readers, and considering it in the differential diagnosis of brain abscess in Sickle Cell disease is a significant crucial point.

Author Response

Comments 1: This is a well-written case report. They described a Salmonella brain abscess in Sickle Cell disease, and as stated in the study, this is the second description in the literature. They believe the mechanism is hematogenous spread. They provided their management strategies, as well as post-operative and pre-operative imaging for their case. I believe this is a useful case report for readers, and considering it in the differential diagnosis of brain abscess in Sickle Cell disease is a significant crucial point.

Response 1: Thank you for your kind comments!